# Stress granules counteract senescence by sequestration of PAI-1

Amr Omer[1], Devang Patel[1], Xian Jin Lian[1], Jason Sadek[1], Sergio Di Marco[1], Arnim Pause[1], Myriam Gorospe[2] & Imed Eddine Gallouzi[1,3,*] (iD)

## Abstract

**Cellular senescence is a physiological response by which an organism halts the proliferation of potentially harmful and damaged cells. However, the accumulation of senescent cells over time can become deleterious leading to diseases and physiological decline. Our data reveal a novel interplay between senescence and the stress response that affects both the progression of senescence and the behavior of senescent cells. We show that constitutive exposure to stress induces the formation of stress granules (SGs) in proliferative and presenescent cells, but not in fully senescent cells. Stress granule assembly alone is sufficient to decrease the number of senescent cells without affecting the expression of bona fide senescence markers. SG-mediated inhibition of senescence is associated with the recruitment of the plasminogen activator inhibitor-1 (PAI-1), a known promoter of senescence, to these entities. PAI-1 localization to SGs increases the translocation of cyclin D1 to the nucleus, promotes RB phosphorylation, and maintains a proliferative, non-senescent state. Together, our data indicate that SGs may be targets of intervention to modulate senescence in order to impair or prevent its deleterious effects.**

**Keywords** PAI-1; senescence; stress granules; stress response
**Subject Categories** Ageing; Physiology

## Introduction

Cellular senescence, a process of irreversible cell cycle arrest, is induced in response to various physiological and environmental stresses to prevent the proliferation of aberrant and potentially harmful cells [1–7]. Senescence-inducing signals such as the DNA-damage response (DDR) and oxidative stress (OS) usually engage the tumor suppressor p53 (TP53) and associated cyclin-dependent kinase inhibitors (CDKIs) p16^Ink4a (CDKN2A) and/or p21 (CDKN1A) [3,5,7,8]. The activation of these two CDKIs establishes senescence

mainly by interfering with the phosphorylation of the retinoblastoma (RB) tumor suppressor protein leading to its inactivation and consequently to cell cycle arrest [4,9]. Additionally, p53 triggers senescence by promoting the expression and secretion of the serine protease inhibitor plasminogen activator inhibitor-1 (PAI-1) which, via an autocrine function, prevents the cyclin D1-dependent phosphorylation of Rb leading to cell growth arrest [5,8,10–13]. These p53-dependent mechanisms, mediated by CDKIs and RB, can act either separately or in combination depending on the stress which induces senescence.

The physiological impact of senescence is complex, as it can lead to beneficial or deleterious outcomes in a context-dependent manner [1,5,14–16]. These opposing effects are mediated by senescence-associated traits including the senescence-associated secretory phenotype (SASP) [17]. The SASP is characterized by the secretion of factors including growth hormones, cytokines, angiogenic factors, and extracellular matrix-remodeling proteases (e.g. GROa1, IL-6, IL-8, and PAI-1) that affect tissues locally and systemically, thereby accelerating or suppressing cancer development or other age-associated conditions like atherosclerosis [1,7,17,18]. Recent observations have established a strong correlation between the increased secretion of PAI-1 and accelerated aging in mice [11], further underscoring the notion that age and stress accelerate the accumulation of senescent cells, in turn affecting the fitness of an organism [1,18]. Although the effect of stress on the induction of senescence and on the SASPs is established, the molecular mechanisms through which it does so remain unclear.

It is well known that some senescent cells escape elimination and remain hidden within tissues for decades where they maintain an active and distinct metabolic profile, adapting and responding to environmental changes and assaults during the lifespan of the organism [1,6,7,19–22]. The abilities to live for a long time and adapt to diverse conditions suggest that senescent cells could exhibit altered phenotypes resulting from repeated exposure to persisting stress conditions. However, the impact of chronic, persistent stress on the senescence process and on the behavior of senescent cells has yet to be explored.

One mechanism by which cells respond to extracellular stresses is through the formation of stress granules (SGs) [23–29]. Work

1 Department of Biochemistry, Rosalind and Morris Goodman Cancer Centre, McGill University, Montreal, QC, Canada
2 Laboratory of Genetics and Genomics, National Institute on Aging-Intramural Research Program, NIH, Baltimore, MD, USA
3 Life Sciences Division, Hamad Bin Khalifa University (HBKU), Education City, Doha, Qatar
*Corresponding author. Tel: +1 514 398 4537; Fax: 1 514 398 7384; E-mails: imed.gallouzi@mcgill.ca or igallouzi@kbku.edu.qa

from our laboratory has shown that senescent cells are able to form SGs in response to oxidative stress such as arsenite [20]. SGs are cytoplasmic entities composed of mRNA and RNA-binding proteins (RBP) and are responsible for the post-transcriptional regulation of genes that modulate various cell processes, some of which are known to affect senescence [20,23–25,30]. The formation of SGs is an important pro-survival mechanism through which cells cope with extracellular stresses by regulating the expression of proteins and providing a means to repair stress-induced damage [23,24,27,28,31,32]. Indeed, numerous mRNAs encoding essential proteins translocate to SGs in association with several RNA-binding proteins (RBPs) such as HuR (human antigen R), TIA-1 (T-cell-restricted intracellular antigen-1), TIAR (TIA-1-related protein), G3BP1 (Ras-GAP SH3 domain-binding protein 1), and FMRP (Fragile X Mental Retardation Protein) [20,32–35]. Since the translocation of many mRNAs to SGs has been correlated with a stress-induced reduction in their translation [28,29], it has been suggested that SGs could play a role in maintaining the recruited mRNAs in a translationally silent state [23,28,29]. More recently, the formation of SGs has been implicated in impacting cell responses through sequestration of key components of well-established signaling pathways such as the NF-κB and p38/JNK pathways [36,37]. Since, as mentioned above, senescent cells form SGs in response to stress [20], the possibility exists that SGs could affect senescence by modulating the activation of signaling pathways that promote cell proliferation such as the cyclin D1/Rb pathway [5,8,10,11].

Although our previous observations established that senescent cells can form SGs in response to acute stress [20], the impact of chronic stress on the formation of SGs and the impact of SGs on senescence are still elusive. Furthermore, the mechanisms by which SGs mediate their effects on senescence also remain unexplored. In this study, we address these questions and show that the repeated exposure to sub-lethal doses of oxidative stress induces the robust formation of SGs during the early stages of senescence. The number of SGs formed during this process, however, decreases as the cells become senescent. Interestingly, our data show that the early formation of SGs in proliferating cells affects their ability to become senescent. Our results suggest that SGs could mediate this effect through the recruitment of PAI-1 to SGs resulting in the sequestration of PAI-1 and the activation of the cyclin D1/Rb pathway. Our data highlight the possibility that SGs could function as a means to interfere with the senescence process to alleviate some of its deleterious outcomes through the life span of an organism.

## Results

### Repeated exposure to oxidative stress impairs the senescence process and prevents the ability of senescent cells to form stress granules

To determine the effect of repeated exposure to stress on senescence, we used two well-established cell models for this process, IDH4 and WI-38 human diploid fibroblasts [38–41]. Senescence was induced in IDH4 cells by removing dexamethasone from the growth media which leads, as previously shown [38,39], and confirmed by our Western blot analysis (Appendix Fig S1A), to a complete inhibition of SV40 T-antigen expression. On the other hand, WI-38 cells, which are primary human fetal lung fibroblasts [17,42], were rendered senescent by a short exposure to ionizing radiation (10 Gy), a treatment known to trigger DNA-damage responses that leads to a permanent growth arrest [17]. Karyotyping analysis on WI-38 fibroblasts used in our experiments did not reveal any obvious chromosomal abnormalities (Appendix Fig S1B). As both cell lines were progressing toward senescence, they were treated daily with a sub-lethal dose of sodium arsenite (AS) (0.5 mM) for 30 min, a well-characterized oxidative stress-inducing agent, and a promoter of SG assembly [30,32]. The effect of stress on the induction of senescence was monitored by measuring the activity of senescence-associated β-galactosidase, a widely used biomarker of senescent cells [43]. The degree of senescence following staining for β-galactosidase activity is reflected both by the percentage of cells that are stained blue-green and by assessing the intensity of the stain in cells with flattened morphology [43].

Cells treated as described above, with or without AS, were fixed at various stages during senescence; proliferating stage (PRO) between days 0 and 3, when most cells were capable of cell division, presenescent stage (PRE) between days 4 and 6, where between 30% and 70% of cells were capable of dividing, and senescent stage (SEN) between days 7 and 10, where only < 30% of cells are capable of dividing [44]. We observed that following AS treatment, the number of senescent cells in both IDH4 and WI-38 was ~50% of that seen in untreated (UNT) cells during both the PRE and SEN stages of the process (Fig 1A and Appendix Fig S2A). Next, we assessed the impact of repeated exposure to AS on the expression levels of p53, p21$^{cip}$, and p16, a set of factors chosen based on their well-established roles in promoting and/or maintaining senescence [18]. Total cell extracts prepared from IDH4 and WI-38 fibroblasts harvested at the PRO, PRE, and SEN stages, exposed or not to AS, were used to perform Western blot analysis with antibodies against p53, p21$^{Cip1}$, and p16$^{INK4a}$. We observed no significant differences in the expression levels of these factors between UNT and AS-treated cells (Fig 1B and Appendix Fig S2B), suggesting that the reduced senescence seen with repeated AS treatment cannot be associated to changes in the expression levels of these key *bona fide* senescence modulators. In addition, this AS-mediated effect was not due to the activation of apoptosis in these cells, since no caspase-3 cleavage products, a well-established marker of apoptosis-induced cell death [45], was detected at any stages of the senescence process in either the presence or the absence of AS (Fig 1C). While these observations clearly show that AS, a well-known promoter of oxidative stress, interferes with the commitment of cells to the senescence process, the mechanisms behind this effect remains unknown.

It is well established that treating cells with oxidative stress agents such as AS triggers two concomitant events: (i) inhibition of translation initiation through phosphorylation of eIF2α and (ii) formation of SGs [20,30,32]. We observed that upon treatment with AS, the levels of phosphorylated eIF2α (p-eIF2α) were elevated at all stages of senescence in both IDH4 and WI-38 when compared to UNT counterparts (Fig 2A and Appendix Fig S3A). Additionally, we also observed that 30 min of exposure to AS was sufficient to completely inhibit general translation in human fibroblasts (Fig 2B). Subsequently, we monitored the formation of SGs in both IDH4 and WI-38 fibroblasts undergoing senescence and treated or not with AS as described above. These cells were fixed at the PRO, PRE and SEN stages and used for immunofluorescence experiments with

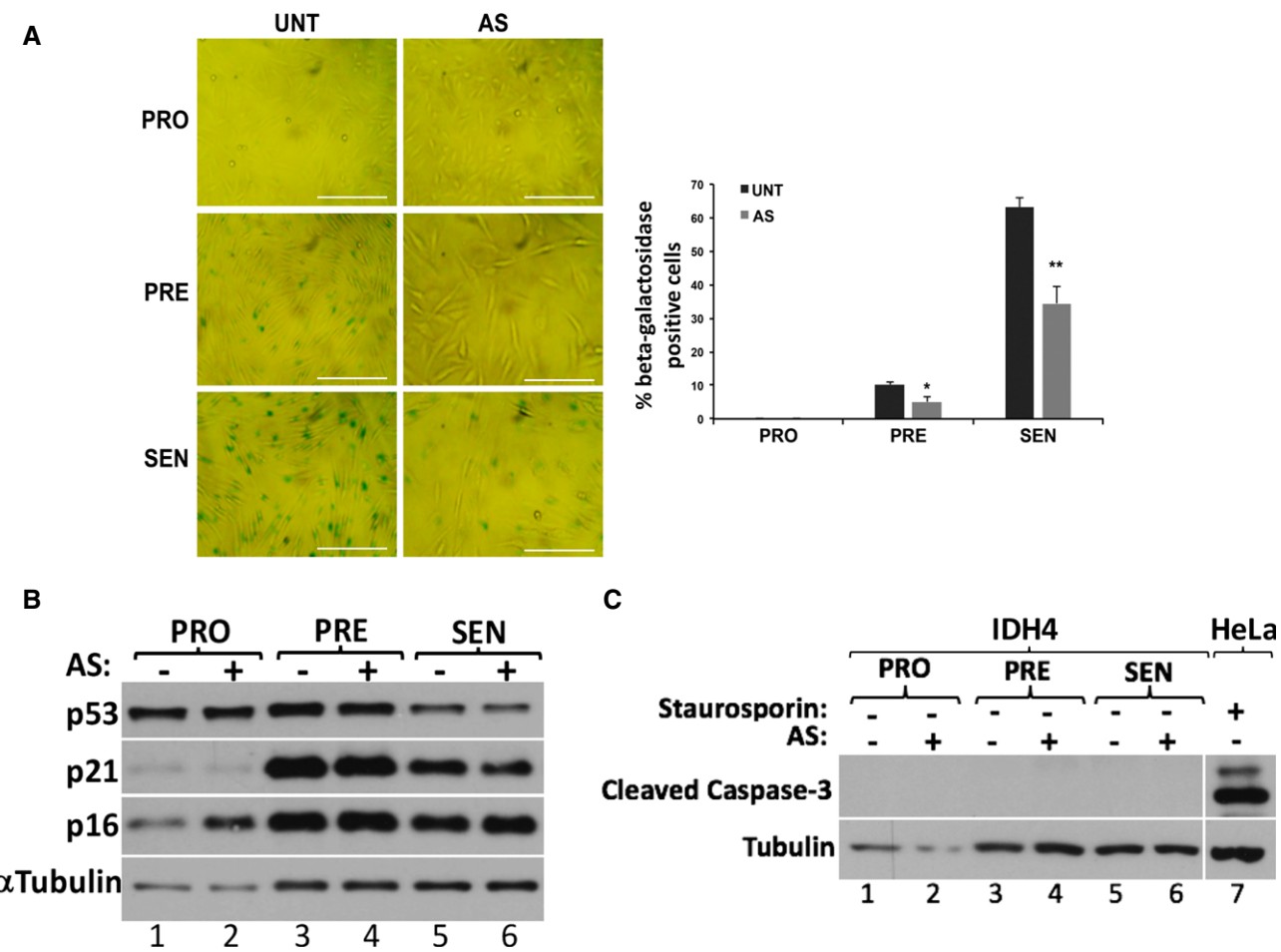

**Figure 1.   Repeated exposure to arsenite decreases the number of cells which commit to the senescence process.**

A   (left) IDH4 cells were treated daily post-induction of senescence for 30 min with (AS) or without (UNT) 0.5 mM sodium arsenite. Proliferating (PRO, Days 0–3), presenescent (PRE, days 4–6) and senescent (SEN, days 7–10) cells were subsequently subjected to staining for β-galactosidase activity. Phase contrast images demonstrate the β-galactosidase staining of the IDH4 cells at the various stages of the senescence process. Scale bars, 400 μm. (right) Graph represents the percentage of cells that stained positive for β-galactosidase activity (stained blue-green) in the phase contrast images shown in (left panel). The percentage of senescent cells in each experiment was calculated using three random fields. Data are represented as a mean of three independent experiments ± SE (error bars). *$P < 0.05$, **$P < 0.01$ (Student's *t*-test).

B   Whole-cell extracts from IDH4 cells treated (AS) or not (UNT) with sodium arsenite as indicated above were prepared and analyzed by Western blot using antibodies for p53, p21, p16, and tubulin (loading control).

C   Whole-cell extracts from IDH4 cells treated (AS) or not (UNT) with sodium arsenite as indicated above were prepared and analyzed by Western blot using antibodies that detect caspase-3 cleavage products and tubulin (loading control). HeLa cells treated with 1 μM staurosporine for 3 h were used as a positive control for caspase-3 cleavage (lane 7).

Source data are available online for this figure.

antibodies against two well-established markers of SGs: FMRP [32,46] and G3BP1 [33]. Although a significant number of SGs formed in PRO cells, the number of SGs in PRE and SEN cells decreased by ~2- and 4-fold, respectively, despite the sustained phosphorylation of eIF2α (Fig 2C and Appendix Fig S3B). Further analysis using immunofluorescence indicated that, under these conditions, β-galactosidase-positive cells have a decreased number of SGs when compared to β-galactosidase-negative cells from the same field (Fig EV1). It is important to note, however, that following each AS treatment, cells were allowed to recover from the stress overnight, prior to the next treatment. During the recovery period, we observed that SGs completely disassembled after 24 hours.

These cells, however, during the PRO and PRE, but not, the SEN stages, re-assembled SGs with subsequent AS treatments (Appendix Fig S4). Therefore, together the data described here clearly indicate that, during senescence, repeated exposure to oxidative stress promotes SG formation at early (PRO and PRE) but not late (SEN) stages of senescence.

**SG formation in the proliferative stage prevents cell commitment to senescence**

The results presented above raise the possibility that the inhibitory effect of AS on senescence could be the consequence of the

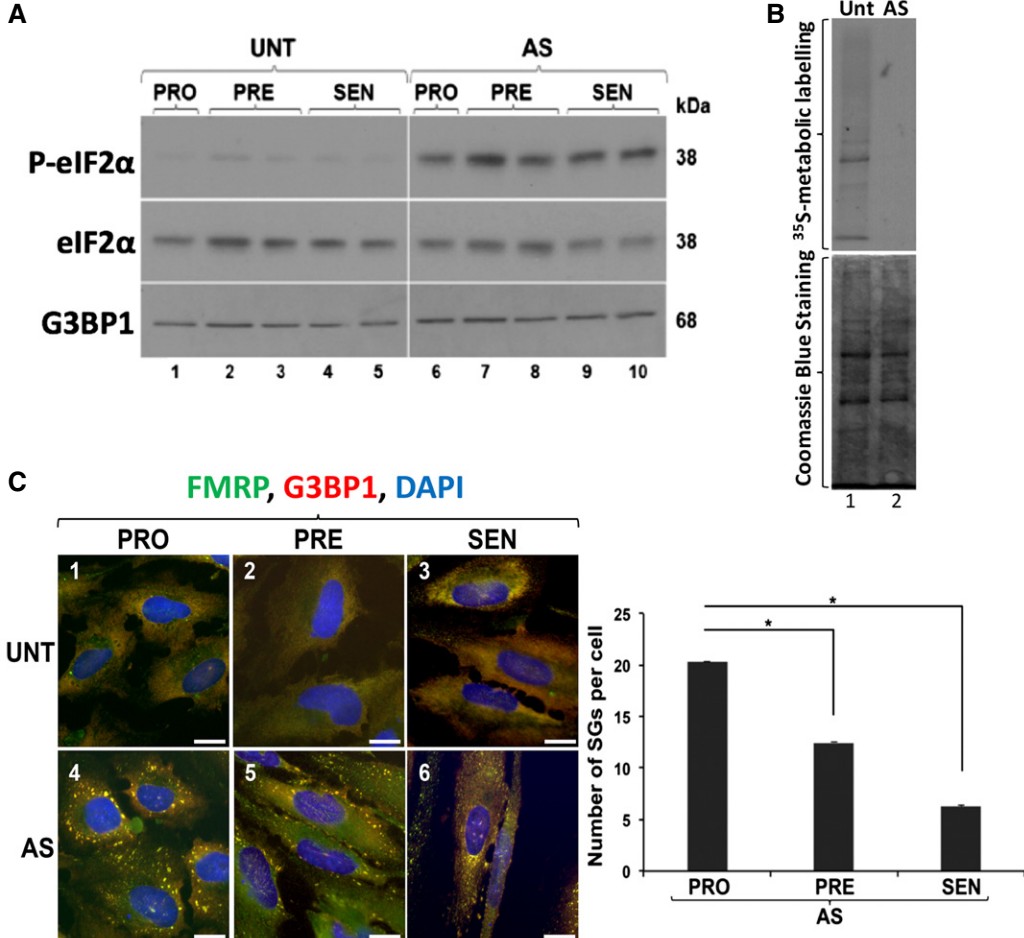

**Figure 2. The formation of stress granules decreases overtime in IDH4 cells exposed to arsenite despite phosphorylation of eIF2α.**

A   Whole-cell extracts from IDH4 cells treated daily post-induction of senescence for 30 min with (AS) or without (UNT) 0.5 mM sodium arsenite were prepared and analyzed by Western blot using antibodies for P-eIF2α, eIF2α, and G3BP1 (loading control).

B   Whole-cell extracts from IDH4 cells treated (AS) or not (UNT) with AS after incorporation of $^{35}$S-methionine were run on SDS–PAGE and visualized by autoradiography or Coomassie Blue staining.

C   (left) After treatment, cells at different stages of senescence [proliferating (PRO), presenescent (PRE), and senescent (SEN)] were fixed, permeablized, and analyzed by immunofluorescence with antibodies against the SG markers FMRP and G3BP1. DAPI staining was used to visualize the nuclei of the cells. Scale bars, 20 μm. (right) Graph represents the number of SGs in the immunofluorescence images (left panel). The average number of SGs per cell was calculated by normalizing the total number of SGs in each field to the total number of cells. Three random fields were used for each quantification. Data are represented as a mean of three independent experiments ± SE (error bars). *$P < 0.05$ (Student's $t$-test).

Source data are available online for this figure.

formation of SGs during the early stages of the process. To investigate this possibility, we determined if daily treatment of IDH4 or WI-38 fibroblasts with AS for three consecutive days only (period during which SGs are most present, Fig 2C and Appendix Fig S3B) could have effects similar to those seen when cells are exposed to AS over 10 days (Fig 1A and Appendix Fig S2A). Interestingly, AS treatment of IDH4 or WI-38 fibroblasts during the PRO stage (3 first days of the senescence process) was sufficient to inhibit senescence, similar to what was seen following 10 consecutive days of treatment (Appendix Fig S5).

We subsequently determined the impact of preventing SG formation in our AS-treated human fibroblasts on the onset of senescence. It is well known that the RNA-binding protein (RBP) Ras-GAP SH3 domain-binding protein (G3BP1), an endoribonuclease, is involved in mediating the assembly of SGs [33,34,47–49]. Our experiments showed that G3BP1 is indeed important for the assembly of SGs in our cells since depleting proliferating fibroblasts of endogenous G3BP1 using siRNA resulted in a significant decrease (~5-fold) in the number of SGs when compared to control cells (Fig 3A and B). These cells were then induced to senesce in the presence or absence of AS stress. Assessment of β-galactosidase activity showed that although the depletion of G3BP1 did not prevent senescence in untreated human fibroblasts, the absence of SGs in these cells allowed them to enter senescence despite repeated exposure to 0.5 mM AS (Fig 3C). The importance of SGs in preventing senescence was further confirmed by the use of cycloheximide (CHX), a

chemical inhibitor of translation elongation and a well-characterized *bona fide* inhibitor of SG formation [20,32,50,51]. Given that the doses of CHX previously used to prevent SG formation (~100 μg/ml) also affect general translation to levels that are similar to AS treatment (Fig 2B) [32,49,51–53], we first determined the minimum dose of CHX that could prevent SG assembly without affecting the levels of newly synthesized proteins. We observed that 0.5 μg/ml of CHX for 30 min was sufficient to prevent both the formation of SGs (Fig 4A) and the AS-mediated impairment of senescence in human fibroblasts (Fig 4B and Appendix Fig S6). However, treatment of fibroblasts with puromycin (Puro), an inhibitor of translation elongation that has no impact on the assembly of AS-induced SGs [54,55], was not able to prevent SG formation nor did it rescue the effect of AS on senescence (Appendix Fig S7). To further confirm the role of SGs in the inhibition of senescence, we exposed human fibroblasts at the PRO stage of senescence to a single dose of pateamine A (PatA), a natural compound that was previously shown to trigger SG assembly independently of eIF2α phosphorylation [51,56]. Interestingly, this single dose of PatA not only led to sustained formation of SGs throughout the 3 days of the PRO phase (Fig 5A), but it also caused the same impairment in senescence that was seen with repeated exposure to AS. The PatA-mediated inhibition of senescence was also reversed by 0.5 μg/ml CHX (Fig 5B).

The data described above establish the importance of SGs in stress-mediated inhibition of senescence. The possibility, nonetheless, exists that other cell damage-associated processes, such as autophagy, the stress response, and/or proteasome–ubiquitination pathways, could also contribute to the effect of SGs on the senescence process [57–62]. In order to address this possibility, we knocked down essential regulators of the autophagy and stress response pathways, Atg5 and HSP70, respectively [63,64], and impaired proteasome function using MG132, a well-established proteasome inhibitor [65]. In all cases, the impairment of these pathways did not rescue or increase the effects of AS-mediated inhibition of senescence (Appendix Figs S8 and S9). Collectively, these results strongly suggest that inducing SG assembly during the early steps of senescence could be sufficient to prevent the commitment of cells to the senescence process.

## SGs interfere with the PAI-1/cyclin D1/pRB-mediated senescence

The recruitment of key components of signaling pathways to SGs has previously been shown to impact cellular processes such as inflammation and apoptosis [36,37]. To uncover the molecular mechanisms by which the early formation of SGs impairs senescence, we investigated whether this effect could be due to the recruitment of key pro-senescent protein(s) to SGs. To identify these factors, we fractionated untreated as well as AS-treated human fibroblasts into soluble and insoluble fractions and then used mass spectrometry analysis to determine their protein content. Work from several laboratories including ours has indicated that in response to various stresses, insoluble fractions of many cells become enriched in numerous SG components [32,34]. Western blot analysis indicated that the levels of *bona fide* SG markers such as G3BP1 and FMRP were enriched in the insoluble fraction of AS-treated presenescent fibroblasts when compared to untreated fibroblasts (Fig 6A). Mass spectrometry analysis identified 845 proteins in the

insoluble fraction of AS-treated fibroblasts, 50 of which are known components of SGs (Fig 6B and Dataset EV1 and EV2). We also found that 0.27% of the factors enriched in the AS-treated insoluble fraction belong to the p53 pathways and only 1 of them, plasminogen activator inhibitor-1 (PAI-1; Fig 6C and Dataset EV1), was previously associated with senescence. Indeed, PAI-1 has been identified as a member of the SASP and several observations have correlated PAI-1 secretion with the onset of senescence, via its ability to inhibit the cyclin D1 pathway [8,11]. Consistent with this finding, Western blot and immunofluorescence analyses showed that in response to inducers of SG assembly, PAI-1 was not only enriched in the insoluble fraction of human fibroblasts (Fig 6D) but was also recruited to SGs (Figs 6E and EV2, and Appendix Fig S10). Together, these results raise the possibility that one way by which SGs impair the senescent process is by interfering with the pro-senescence function of PAI-1.

Since PAI-1 is a well-known promoter of senescence downstream of p53 [8,11,66], its depletion or chemical inhibition should impact the induction of senescence similarly to when primary fibroblasts are repeatedly exposed to AS. To determine the importance of PAI-1 in this process, we used specific siRNA duplexes to knockdown PAI-1 expression and employed the well-established small-molecule inhibitor of PAI-1 TM5441 (Fig 7A) [12,13,67–69]. Our data indicate that both PAI-1 knockdown (Fig 7B) and TM5441-mediated inhibition (Fig 7C) were sufficient to promote a significant decrease in the number of β-galactosidase-positive cells in the presence or absence of AS treatment. Next, we assessed the impact of repeated exposure to AS on PAI-1 secretion. ELISA experiments using supernatants from fibroblasts induced into senescence showed a significant reduction in PAI-1 secretion in response to repeated exposure to AS; this AS-elicited suppression is rescued by treatment with a dose of CHX that impairs the formation of SG (Fig 8A and Appendix Fig S11A). Knocking down G3BP1 reestablished PAI-1 secretion in fibroblasts induced into senescence exposed to chronic treatment of AS (Appendix Fig S11B and C). In addition, our data showed that PatA also significantly reduced the secretion of PAI-1 and that this effect was reversed by 0.5 μg/ml CHX (Appendix Fig S11D and E). These observations indicate that the translocation of PAI-1 to SGs in response to AS could contribute to the SG-elicited repression of senescence by interfering with the secretion of PAI-1. Based on these findings, we hypothesized that the overexpression of PAI-1 in fibroblasts or the supplementation of these cells with recombinant exogenous PAI-1 should reverse the AS-mediated effects on senescence. Further experiments confirmed this assumption and showed that both PAI-1 overexpression and supplementation with recombinant human PAI-1 reversed the senescence inhibition of human fibroblasts that was seen with AS or PatA treatment (Figs 8B, EV3 and EV4, and Appendix Fig S12).

It was previously shown that secreted PAI-1 induced senescence by promoting the cytoplasmic retention of cyclin D1, which in turn prevented the phosphorylation of pRb leading to cell cycle arrest [8,70]. Assessment of the impact of AS on this PAI-1-regulated pathway by immunofluorescence and Western blot experiments showed that AS promoted the translocation of cyclin D1 to the nucleus, which in turn led to a significant increase in the phosphorylation levels of Rb (Fig 8C and D, and Appendix Fig S13). Together, our results suggest that SGs prevent the commitment of cells to

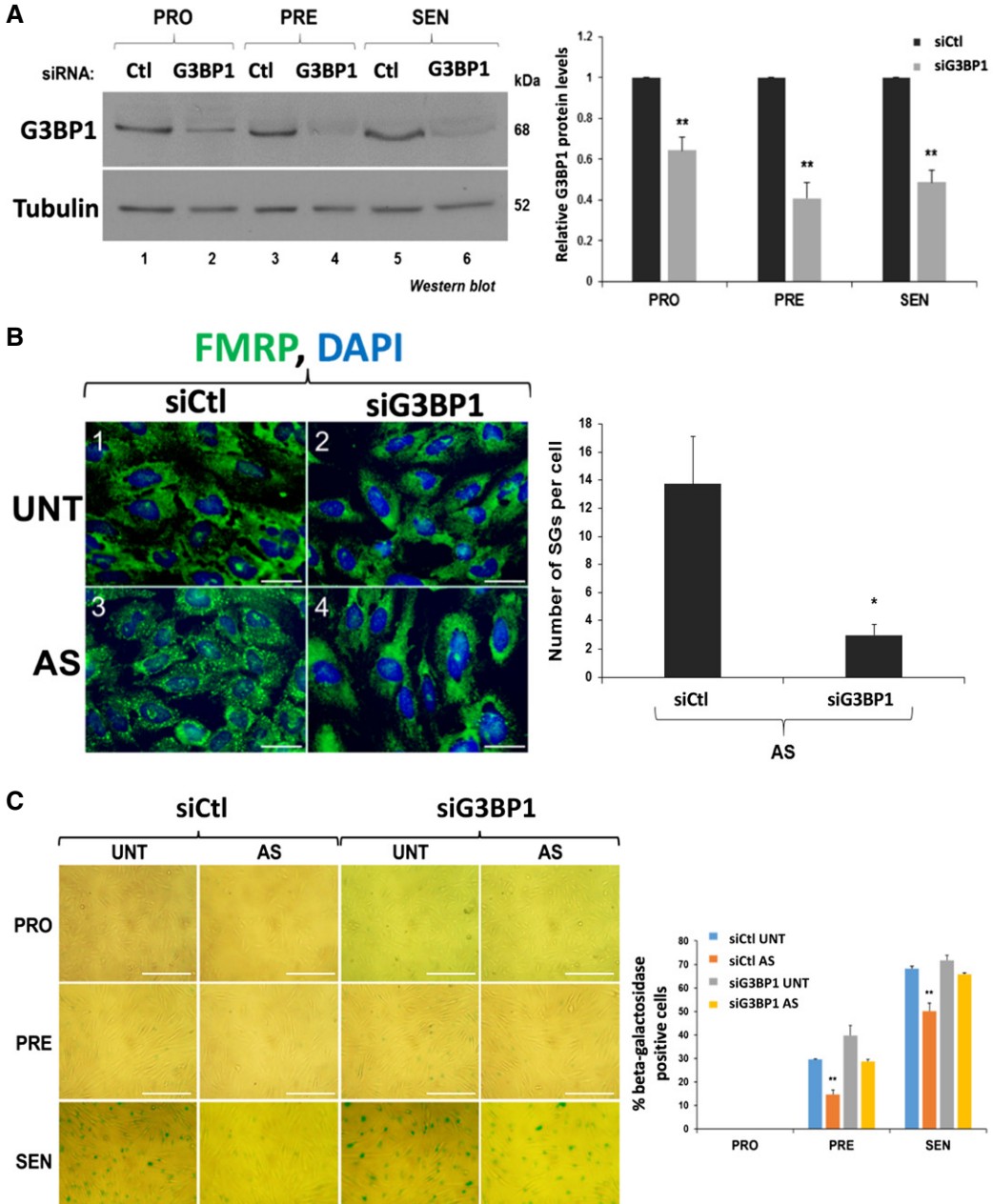

**Figure 3. Knockdown of G3BP1 prevents the assembly of stress granules and the arsenite-mediated effect of SGs on senescence.**

A   (left) Proliferating IDH4 cells were transfected with a control (Ctl) or a G3BP1-specific siRNA, and senescence was induced 24 h post-transfection. Western blots were performed using whole-cell extracts from PRO, PRE, and SEN cells, and antibodies specific for G3BP1 and tubulin (loading control) proteins. (right) ImageJ was used to quantify the levels of G3BP1 protein, which were normalized to those of tubulin protein. The graph represents the normalized G3BP1 protein levels in the siG3BP1 condition relative to siCtl in PRO, PRE, and SEN cells.

B   (left) IDH4 cells, transfected with siRNA targeting G3BP1, as in (A), were treated daily for 30 min with 0.5 mM sodium arsenite (AS). Cells at the PRO stage were subsequently fixed, permeabilized, and analyzed by immunofluorescence with an antibody specific for FMRP protein. DAPI staining was used to visualize the nuclei of the cells. Scale bars, 40 μm. (right) Graph represents the number of SGs in proliferating IDH4 cells shown in left panel. The average number of SGs per cell was calculated by normalizing the total number of SGs in each field to the total number of cells. Three random fields were used for the calculation.

C   (left) IDH4 cells transfected as in (A) were treated daily during senescence for 30 min with (AS) or without (UNT) 0.5 mM sodium arsenite. Cells at the PRO, PRE, and SEN stages were subsequently subjected to staining for β-galactosidase activity. Phase contrast images demonstrating β-galactosidase staining are shown. Scale bars, 400 μm. (right) Graph represents the percentage of cells that stained positive for β-galactosidase activity (stained blue-green in left panel). The percentage of senescent cells in each experiment was calculated using three random fields.

Data Information: (A, B, and C) Data are represented as a mean of three independent experiments ± SE (error bars). *P < 0.05, **P < 0.01 (Student's t-test).
Source data are available online for this figure.

    

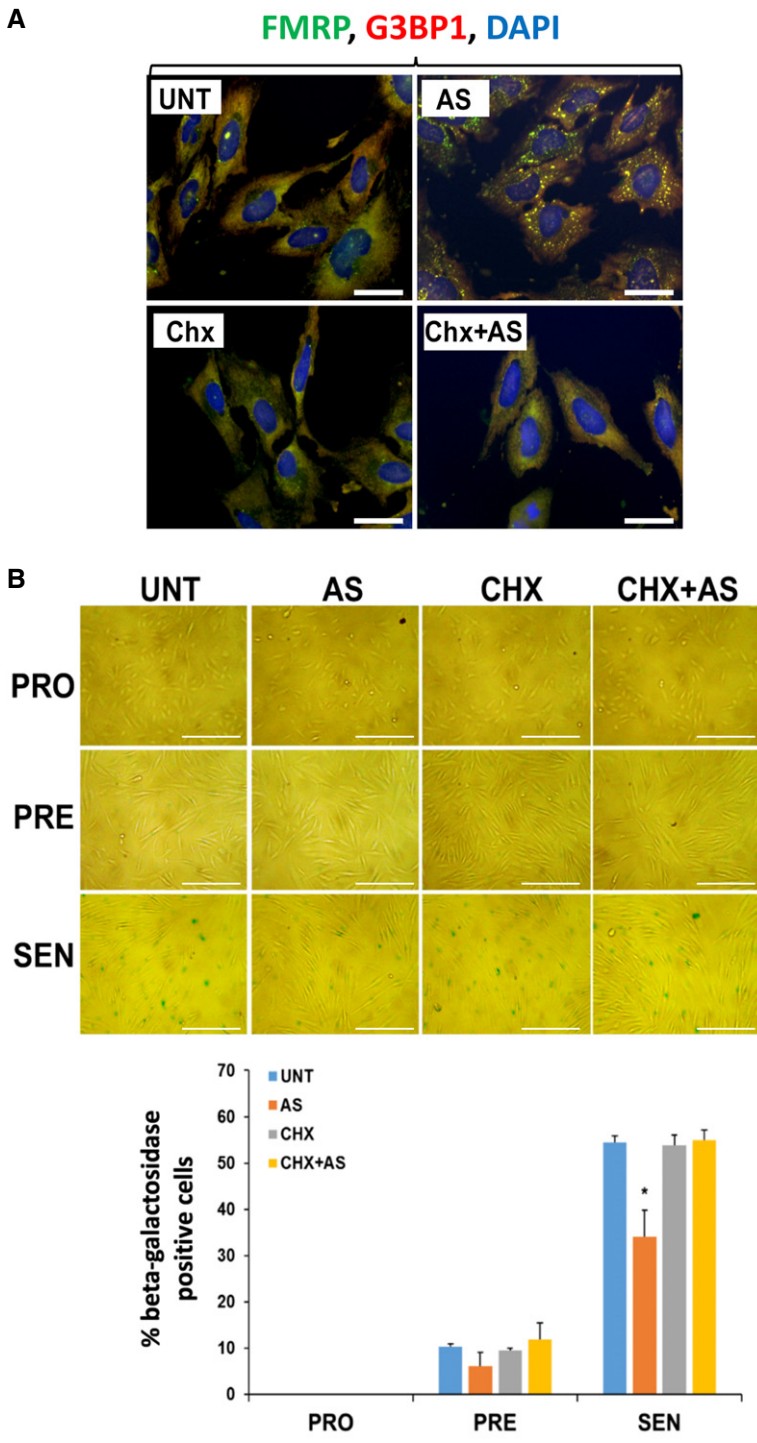

**Figure 4.  Cycloheximide inhibits arsenite-induced stress granule formation and prevents the decreased number of senescent cells mediated by repeated arsenite treatment.**

A   Proliferating IDH4 cells were treated for 30 min with (AS) or without (UNT) 0.5 mM sodium arsenite, 0.5 µg/ml cycloheximide (CHX), or both 0.5 mM sodium arsenite and 0.5 µg/ml cycloheximide (CHX + AS). Cells were subsequently fixed, permeablized, and analyzed by immunofluorescence with antibodies against G3BP1 and FMRP proteins. DAPI staining was used to visualize the nuclei of the cells. Scale bars, 40 µm.

B   (top) IDH4 cells were treated daily during senescence for 30 min with or without AS and/or CHX as in (A). Cells at the PRO, PRE, and SEN stages were subsequently subjected to staining for β-galactosidase activity. Phase contrast images demonstrating the β-galactosidase staining of the IDH4 cells at the various stages of the senescence process. Scale bars, 400 µm. (bottom) Graph represents the percentage of cells that stained positive for β-galactosidase activity (stained blue-green) in (top panel). The percentage of senescent cells in each experiment was calculated using three random fields. Data are represented as a mean of three independent experiments ± SE (error bars). *$P < 0.05$ (Student's $t$-test).

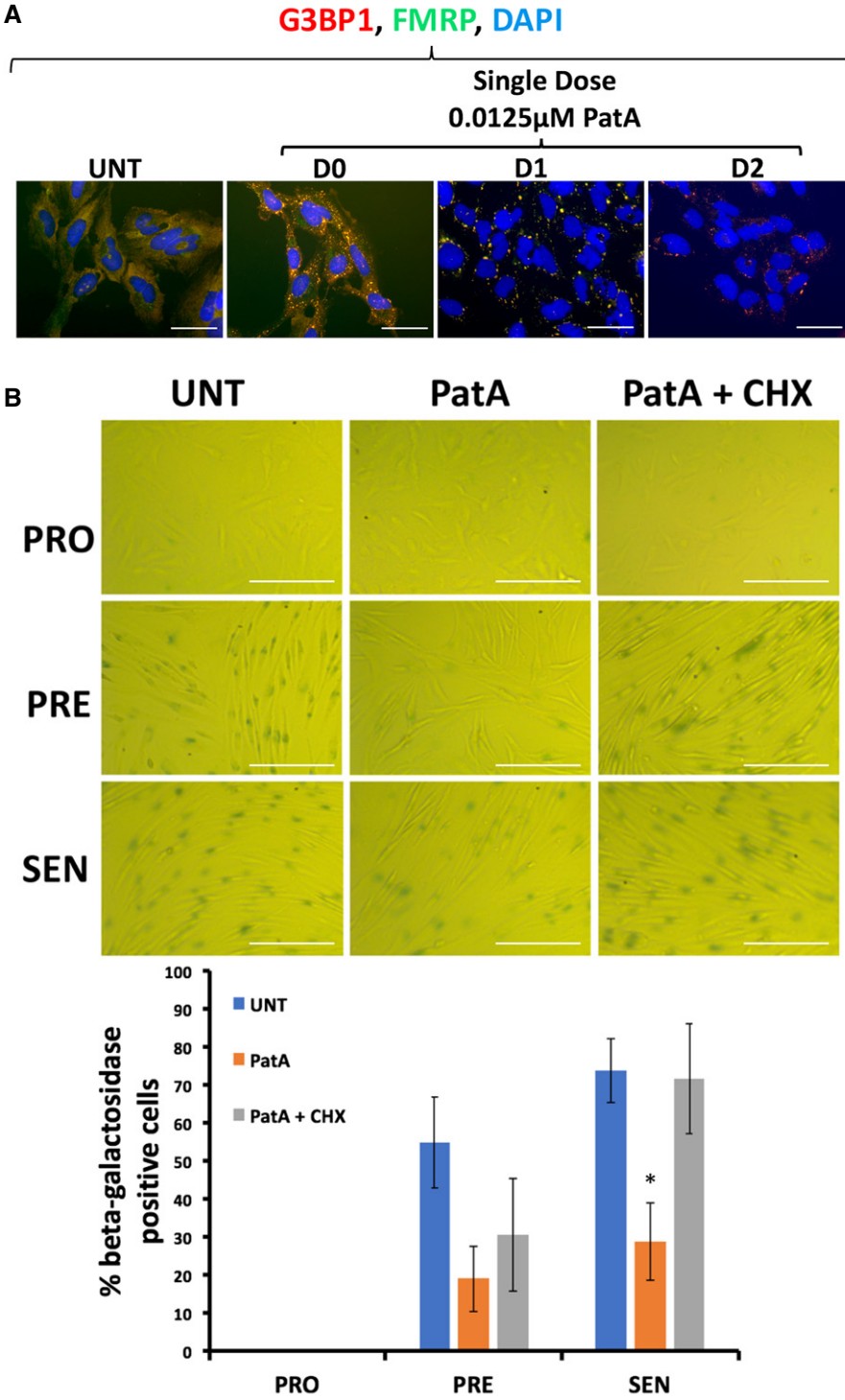

**Figure 5. Exposure to pateamine A leads to formation of stress granules and decreases the number fibroblasts that become senescent.**
IDH4 cells were treated with a single dose of 0.0125 μM pateamine A (PatA) for 30 min.

A   Cells were fixed at days 0, 1, and 2, permeabilized and analyzed by immunofluorescence using antibodies against G3BP1 and FMRP. Scale bars, 50 μm.

B   (top) IDH4 cells were treated post-induction of senescence for 30 min with or without 0.0125 μM PatA. PRO, PRE, and SEN cells were subsequently subjected to staining for β-galactosidase activity. Phase contrast images demonstrating the β-galactosidase staining of the IDH4 cells at the various stages of the senescence process. Scale bars, 200 μm. (bottom) Graph represents the percentage of cells that stained positive for β-galactosidase activity (stained blue-green) in the phase contrast images shown in the top panel. The percentage of senescent cells in each experiment was calculated using three random fields. Data are represented as a mean of three independent experiments ± SE (error bars). *$P < 0.05$ (Student's *t*-test).

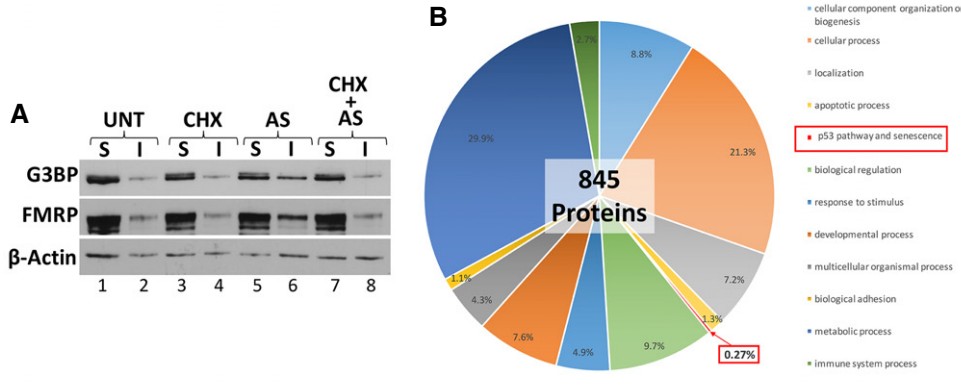

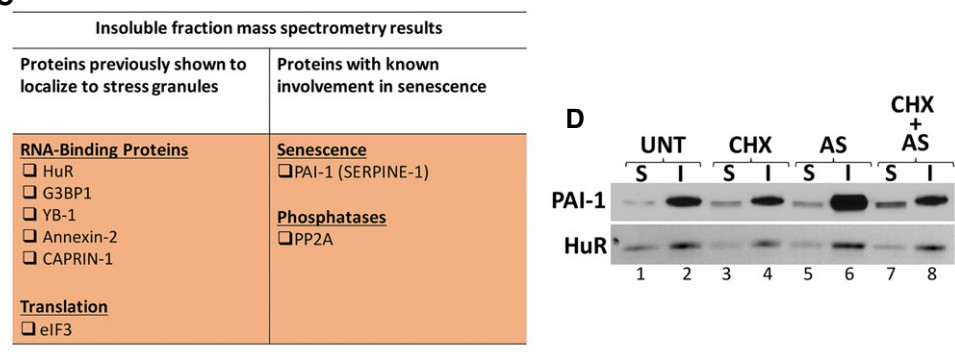

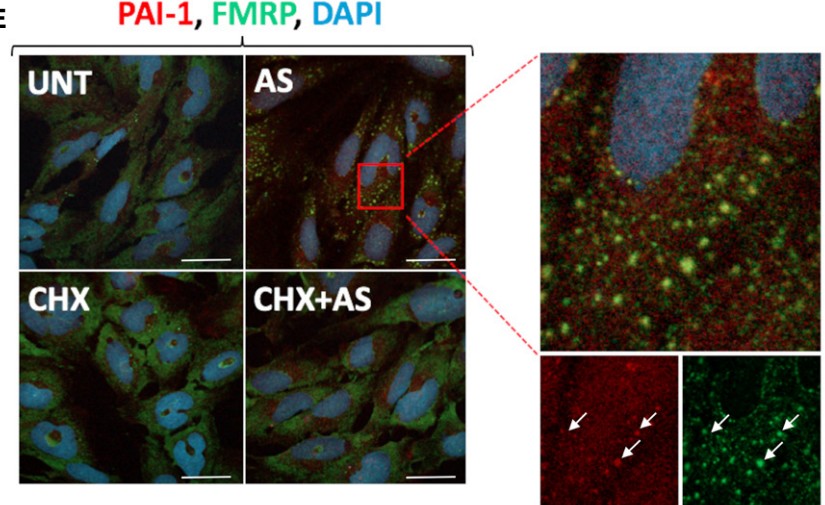

**Figure 6.  Mass Spectrometry analysis reveals PAI-1 as a novel component of stress granules.**

A   IDH4 cells treated with arsenite and/or cycloheximide (CHX) for thirty minutes and harvested at the PRO stage were lysed and fractionated into soluble (Sol) and insoluble (Ins) fractions. Proteins in each fraction were resolved on a 10% SDS–polyacrylamide gel, and analyzed by Western blot. Western blots were performed using antibodies specific for G3BP1, FMRP, and β-actin (loading control) proteins.

B   Mass spectrometry analysis of proteins found in arsenite-treated insoluble fractions separated by biological function.

C   Table displaying sample list of proteins found in (B), and factors involved in senescence.

D   Soluble and insoluble fractions were resolved on a 10% SDS–polyacrylamide gel [as in (A)], and analyzed by Western blot. Western blots were performed using fractions from PRO cells, and antibodies specific for PAI-1 and HuR (positive control) proteins.

E   Cells at the PRO stage were fixed, permeabilized, and analyzed by immunofluorescence with an antibody specific for FMRP and PAI-1 proteins. DAPI staining was used to visualize the nuclei of the cells. The red square in the AS panel represents the area that was expanded and is shown on the right. The two panels below the expanded box show individual staining for PAI-1 (red) and FMRP (green). Arrows indicate examples of co-localized PAI-1 and FMRP in the same foci. Scale bars, 50 μm.

Source data are available online for this figure.

                                                                      

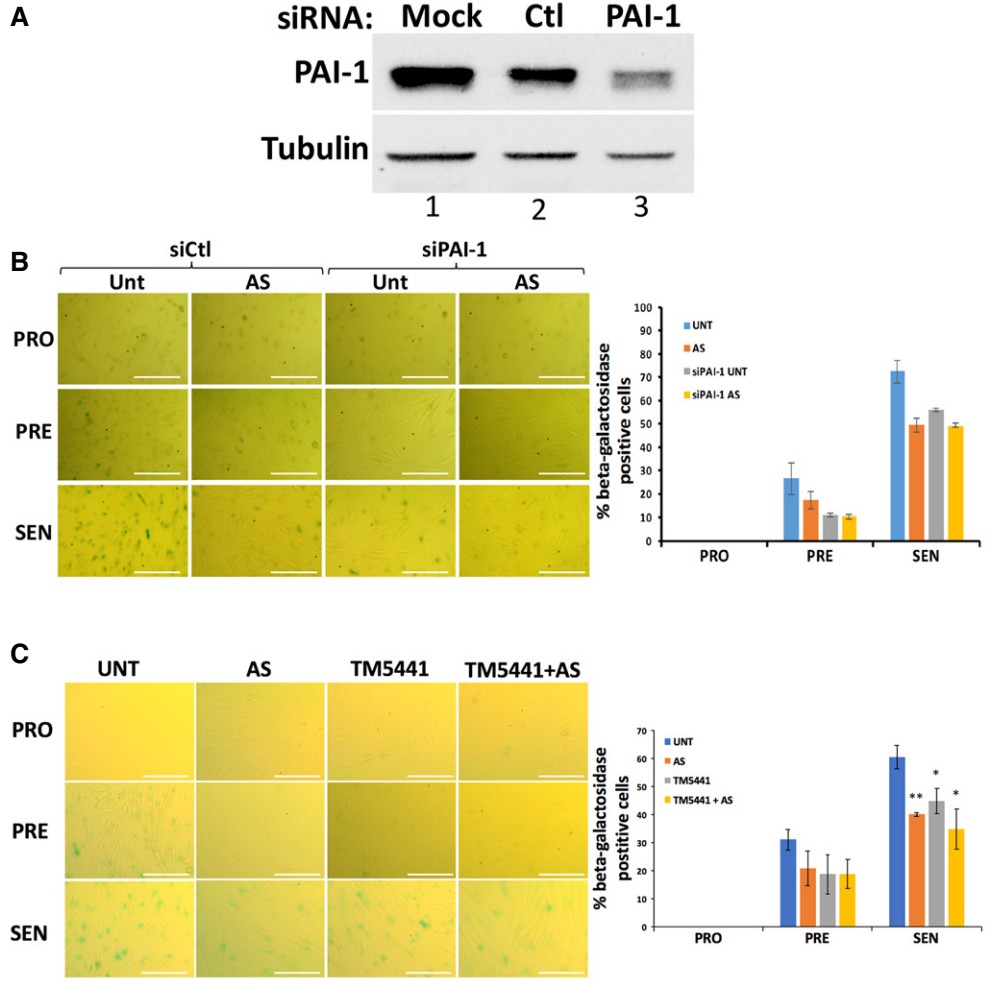

**Figure 7. PAI-1 depletion or chemical inhibition decreases the number of fibroblasts that become senescent.**

A   Proliferating IDH4 cells were transfected with a control (Ctl) or a PAI-1-specific siRNA, and senescence was induced 24 h post-transfection. Western blots were performed using whole-cell extracts from cells at the PRO stage, and antibodies specific for PAI-1 and tubulin (loading control) proteins.

B   (left) IDH4 cells transfected as in (A) were treated daily during senescence for 30 min with (AS) or without (UNT) 0.5 mM sodium arsenite. Cells at the PRO, PRE, and SEN stages were subsequently subjected to staining for β-galactosidase activity. Phase contrast images demonstrating β-galactosidase staining are shown. Scale bars, 200 μm. (right) Graph represents the percentage of cells that stained positive for β-galactosidase activity (stained blue-green) in (left panel). The percentage of senescent cells in each experiment was calculated using three random fields.

C   (left) IDH4 cells were treated daily during senescence for 30 min with or without arsenite (AS). These cells were simultaneously also treated with or without 10 μM of TM5441 for 24 h daily after treatment with AS. Cells at the PRO, PRE, and SEN stages were subsequently subjected to staining for β-galactosidase activity. Phase contrast images demonstrating β-galactosidase staining are shown. Scale bars, 200 μm. (right) Graph represents the percentage of cells that stained positive for β-galactosidase activity (stained blue-green) in (left). The percentage of senescent cells in each experiment was calculated using three random fields.

Data Information: (B and C) Data are represented as a mean of three independent experiments ± SE (error bars). *$P < 0.05$, **$P < 0.01$ (Student's *t*-test).
Source data are available online for this figure.

senescence by interfering with PAI-1 secretion and with the ability of PAI-1 to inhibit the cyclin D1/pRB pathway.

## Discussion

Many studies have shown that the senescence phenotype is elicited in response to various stresses. However, unlike apoptotic or necrotic cells, which lead to complete cell collapse and rapid elimination, senescent cells resist cell death, maintain an active

metabolism, and are able to persist and accumulate over an extended period of time [2,4,71,72]. Moreover, although senescence is induced by various stresses, cells can still respond and adapt to subsequent environmental challenges [20]. Despite this phenotypic plasticity, little is known regarding the impact of repeated exposure to stress on the fate and behavior of senescent cells. In this study, we address this gap and report that, while frequent stresses can prevent senescence, the senescent state itself affects the assembly of SGs, one of the well-established cell stress response mechanism. We observed that treating cells induced into senescence cells daily with

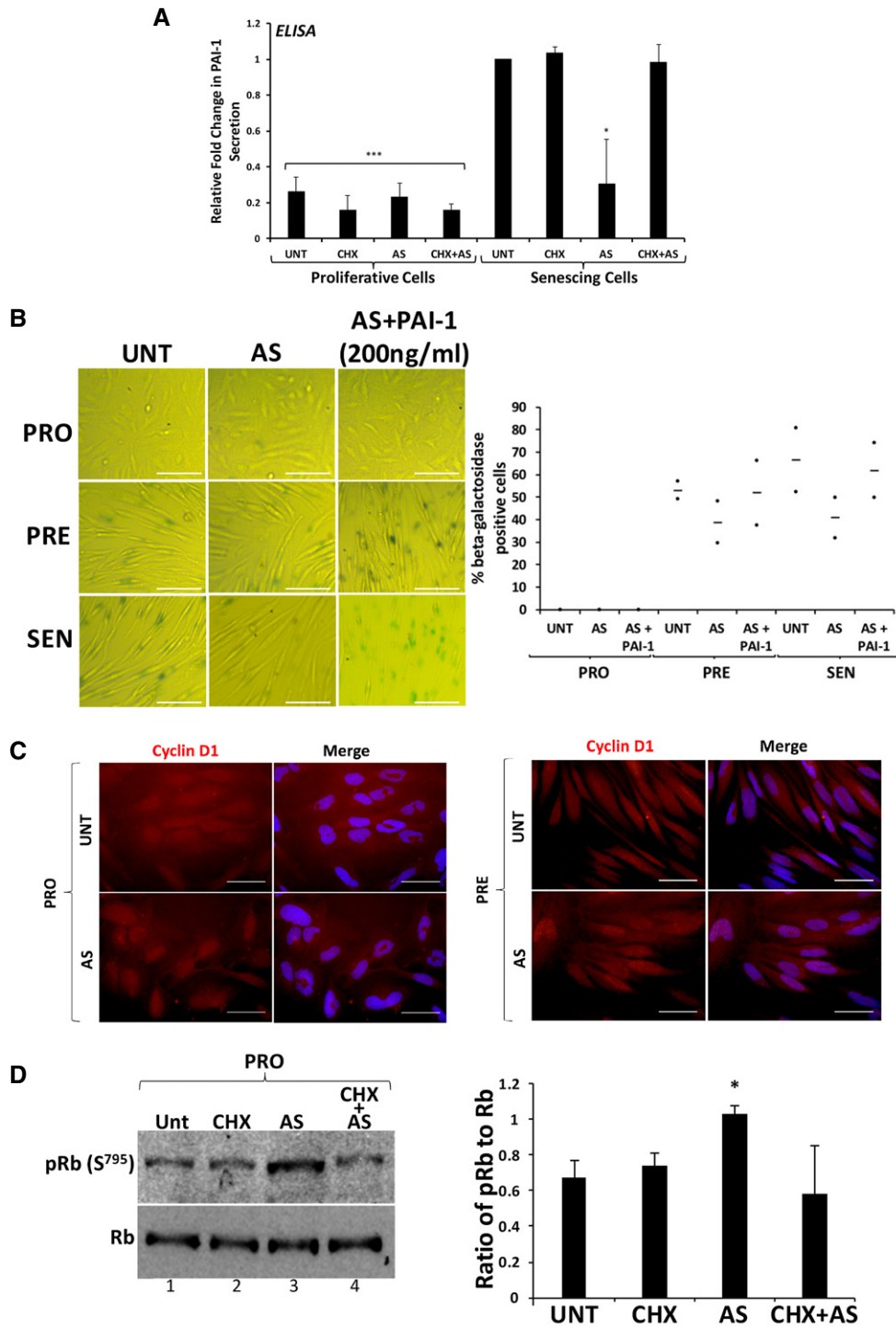

**Figure 8.**

a sub-lethal dose of oxidative stress triggers the formation of SGs during the early steps of the process but prevents SG assembly in fully senescent cells. Interestingly, the early induction of SGs was sufficient to trigger a significant decrease in the number of proliferating cells that become senescent. Mechanistically, we found that the SG-mediated inhibition of senescence correlates with their ability to recruit PAI-1, a member of the SASP and a well-known promoter of senescence [11–13,69]. Together, our results support a model whereby the formation of SGs during the early steps of senescence is sufficient to suppress the commitment of cells to irreversible growth arrest. One way by which SGs could mediate this effect is by recruiting PAI-1 and by preventing its secretion as well

**Figure 8.  The recruitment of PAI-1 to stress granules interferes with its secretion resulting in the activation of the cyclin D1/RB pathway.**

A   Secreted PAI-1 levels in supernatant obtained from senescing IDH4 cells treated with AS with or without cycloheximide (CHX) were assessed by ELISA.

B   (left) IDH4 cells treated daily during senescence for 30 min with arsenite (AS) were supplemented or not with 200 ng/ml of recombinant PAI-1. Cells at the PRO, PRE and SEN stages were subsequently subjected to staining for β-galactosidase activity. Phase contrast images demonstrating β-galactosidase staining are shown. Scale bars, 200 μm. (right) Graph represents the percentage of cells that stained positive for β-galactosidase activity (stained blue-green) in (left). The percentage of senescent cells in each experiment was calculated using three random fields.

C   Cells at the PRO (left panels) and PRE (right panels) stages were fixed, permeabilized and analyzed by immunofluorescence with an antibody specific for cyclin D1. Merged panels display DAPI staining and cyclin D1 staining in order to visualize the location of nuclei in cells. Scale bars, 50 μm.

D   (left) Western blot analysis was performed using whole-cell extracts from PRO cells, and antibodies specific for pRb (S795) and Rb proteins. (right) ImageJ was used to quantify the levels of pRb (S795) protein, which were normalized to those of Rb protein. The graph represents the normalized pRb (S795) protein signals in the UNT condition relative to treated cells in PRO.

Data Information: (B) Data are represented as a mean (represented by −) of two independent experiments. (A, D) Data are represented as a mean of three independent experiments ± SE (error bars). *$P < 0.05$, ***$P < 0.001$ (Student's $t$-test).
Source data are available online for this figure.

as its ability to interfere with the cyclin D1/pRB-mediated cell growth.

In response to various stresses, cells initially trigger survival pathways that prevent damage and promote recovery; however, if the assault persists and becomes unsustainable, cells undergo death or enter senescence [5,14,26,73,74]. Our data raise the intriguing possibility that during the initial steps of senescence, cells preserve the ability to activate normal stress response mechanisms such as the formation of SGs. However, this response is lost as the cell commits to the irreversible cell cycle arrest of senescence. While oxidative stress-induced SGs did not have any notable effect on the expression of major senescence-associated proteins, the repeated formation of SGs during the early stages of senescence was sufficient to cause a significant delay in cell commitment to this process (Fig 1A and B, and Appendix Fig S2A and B). Indeed, treating cells with oxidative stress during the first 3 days of the process was sufficient to delay senescence (Appendix Fig S5). Furthermore, direct inhibition of SG formation by knocking down G3BP1 or by treatment with CHX prevented this effect (Figs 3 and 4). In addition, the ability of a single dose of PatA to decrease the number of cells that become senescent during the process further supported the notion that the formation of SGs by itself was sufficient to mediate this inhibitory effect. The effect of PatA on this process, however, was more pronounced than AS treatment. While these observations provide a strong support for the role of SGs as a novel modulator of senescence, we still do not know the molecular mechanism affecting SG dynamics (assembly/disassembly) during senescence. Addressing this issue will not only identify to the factors/processes involved in SG formation in these cells but will also provide insight into the impact that the cellular state (proliferation or senescence) could have on SG dynamics and function.

It is well accepted that one of the main functions of SGs during cell response to stress is to recruit numerous mRNAs to prevent their translation and protect them from decay [26,73,74]. However, recent publications have suggested an expansion on the role of SGs beyond the recruitment of mRNPs and mRNAs, but instead as important gathering site for components of cell signaling pathways during various assaults. For example, the sequestration of the TNF receptor-associated factor 2 (TRAF2) protein to SGs, due to its interaction with the known SG component eIF4GI, was associated with the inhibition of the NF-κB pathway during inflammation [36]. Moreover, recruitment of the signaling scaffold protein RACK1 to SGs was shown to inhibit activation of the apoptotic response by the p38/JNK signaling pathways [37]. Therefore, these observations

suggest a function for SGs during the cellular stress response beyond their proposed role as storage sites for RBPs and associated mRNAs [36,37,73]. Our data further support the notion that the SG-mediated delay of senescence involves the recruitment of a key promoter of senescence, plasminogen activator inhibitor-1 (PAI-1) to SGs (Figs 6 and EV2, and Appendix Fig S10). Using mass spectrometry to analyze cellular fractions enriched for SG components, we discovered PAI-1, a serine protease inhibitor and SASP factor, as a novel SG component. At present, there are no efficient biochemical methods to isolate pure SGs from cells [32]. Although the fractionation method used in our study was crude (containing various proteins from insoluble organelle such as endoplasmic reticulum, Golgi apparatus, mitochondria, and nuclear), it represented a simplified and consistent method to gain enrichment in SG components, allowing the discovery of proteins recruited to SGs. Using methods such as immunofluorescence, Western blot, and ELISA to validate the results of the mass spectrometry analysis, we were able to further decipher the effects of stress (specifically AS and PatA) on PAI-1 secretion and localization to SGs (Figs 6–8 and EV2, and Appendix Figs S11–S13). Impairment of SG formation, through the use of CHX, reestablished the secretion and localization of PAI-1 (Figs 6E and 8A, and Appendix Figs S10 and S11). Overall, our data further expand the role of SGs and show that they can serve as hubs for secreted proteins, a role that do not necessarily involve impact on mRNA translation and/or fate. It will be interesting to determine whether this mRNA-independent effect of SGs could modulate other cellular processes and identify the specific factors in SGs required for the recruitment of signaling proteins to SGs.

Previous studies have shown that PAI-1 is regulated downstream of p53 and is sufficient to induce senescence [11,66]. Furthermore, direct and indirect impairment of PAI-1 was previously shown to impair senescence [12,13,75]. Here, we found evidence that exposure of cells progressing toward senescence to chronic oxidative stress promotes the recruitment of PAI-1 to SGs and that this mobilization is sufficient to interfere with the PAI-1-dependent pathway of cell cycle arrest (Fig 8C and D, and Appendix Fig S13). Additionally, our data show that depleting PAI-1 expression in human fibroblasts undergoing senescence impaired the cell commitment to this process similar to repeated exposure to oxidative stress (Fig 7), while supplementation of PAI-1 reversed the effects of arsenite (Figs 8B and EV3, and Appendix Fig S12). A recent study showed that while overexpression/secretion of PAI-1 led to premature aging in mice, the disruption of PAI-1 production reversed this phenotype [11]. Moreover, various disease states, such as cancer,

cardiovascular disease, and diabetes, are also correlated with increased PAI-1 expression and secretion [68,76]. Therefore, developing ways to interfere with PAI-1 function has been the focus of several groups since the establishment of its role in aging-related diseases. The mechanisms by which PAI-1 is mobilized to SGs are not clear at this time, and further investigation is required to elucidate how the transient retention of PAI-1 in these ephemeral organelles (in the case of AS) affects PAI-1 secretion and impact on senescence.

Overall, our work shows that the induction of SG assembly via various means (e.g., AS or PatA) upon induction of senescence is a novel way by which senescence and its deleterious effects could be controlled. A better understanding of the impact that various stresses have on senescence could help design novel approaches to harness the benefits of the senescence phenotype while preventing its deleterious outcomes.

# Materials and Methods

## Cell lines and cultures

IDH4 cells were provided by Shay [38], and WI-38 cells were generously provided by Dr. C. Autexier (Lady Davis Institute, McGill University, QC, Canada). Both cell lines were grown in a 5% $CO_2$ environment at 37°C in Dulbecco's modified Eagle's medium (DMEM) (Invitrogen) supplemented with 10% fetal bovine serum (FBS) (Sigma) and 1% penicillin/streptomycin antibiotics (Sigma). Proliferation of IDH4 cells was maintained by supplementing the growth media with 1 μM dexamethasone (dex) [38]. Senescence of IDH4 cells was induced by removing dex from the media, and replacing the FBS with charcoal-stripped FBS (J R Scientific). WI-38 cells were induced into senescence by exposure to ionizing radiation (10 Gy) when cells were ~70% confluent. Cells were treated with daily with 0.5 mM of sodium arsenite (AS) (Sigma) for 30 min, or with 0.0125 μM of pateamine A (PatA) (a kind gift from Dr. P. Northcote) once for 30 min on day 0 to induce SG formation. Where indicated, these cells were also treated simultaneously with or without 0.5 μg/ml of cycloheximide (CHX) (Sigma) to prevent AS-induced SG formation, or with or without 2.5 μg/ml of puromycin (Puro) (Sigma, P8833). Cells were washed twice with PBS after each treatment and cultured in fresh media. All cells were collected, assayed, or fixed at distinct phases of senescence: Days 0–3 coincided with the proliferating phase (PRO), days 4–6 were the presenescent phase (PRE), and days 7–10 the senescent phase (SEN).

## β-galactosidase staining

The Senescence Cells Histochemical Staining Kit (Sigma) was used to detect senescent cells as previously described [43] following the manufacturer's protocol.

## siRNA

siRNA specific for G3BP1 (5′-UCAACAUGGCGAAUCUUGGTG-3′; siRNA ID # 126650) was obtained from Ambion. siRNA for PAI-1 was obtained from ThermoFisher Scientific (ID-118572). siRNA specific for Atg5 was obtained from Ambion (5′-AUGAGCUUCAAUUG CAUCCTT-3′; siRNA ID # s18158). siRNA specific for HSP70 was obtained from Santa Cruz Biotechnology (sc-29352). Control siRNA was from Dharmacon (5′-AAGCCAAUUCAUCAGCAAUGG-3′).

## Plasmid

Plasmid containing full-length human PAI-1 with C-terminal GFPSpark tag was obtained from Sino Biological (HG10296-ACG).

## Transfection

IDH4 and WI-38 cells were transfected with siRNA or cDNA encoding GFP-PAI-1 using jetPRIME (Polyplus transfection) following the manufacturer's protocol when the cells reached 50–60% confluency. Cells were induced into senescence 24 h after transfection and samples collected at the proliferating (PRO) stage were treated and collected 48 h after transfection.

## Immunofluorescence

IDH4 and WI-38 cells grown on coverslips were fixed, and immunofluorescence experiments were performed as described [20] using antibodies against FMRP (1/6 dilution, kindly provided by Dr. R. Mazroui, Centre Hospitalier de l'Université Laval, Quebec, Canada), G3BP1 (1/1,000 dilution), PAI-1 (1/250, Santa Cruz), and cyclin D1 (1/50, Abcam). In addition, immunofluorescence was performed to assess β-galactosidase activity using 5-dodecanoylaminofluorescein Di-β-D-galactopyranoside ($C_{12}$FDG) (ThermoFisher, D2893). Cells were treated for 1 h with bafilomycin A1 (Sigma-Aldrich, B1793-2UG), which disrupts lysosome formation leading to diffusion of β-galactosidase into the cytosol, and then with $C_{12}$FDG for 2 h. Before fixation, cells were treated with AS to induce SGs and immunofluorescence was performed using the Ziess Axio Observer.Z1 or the ZEISS LSM 800 Laser Scanning Microscope. The number of SGs was determined using ImageJ as previously described [77].

## Western blot analysis

Western blot was performed using total protein extracts and probing with antibodies against G3BP1 (1/2,000), FMRP (1/6) HuR (3A2) (1/10,000) [35], p53 (1/1,000, Calbiochem), p21 (1/2,000, Santa Cruz Biotechnology), p16 (1/1,000, Santa Cruz Biotechnology), SV40 T-antigen (1/10,Abcam), GFP (1/1,000, Clontech), phospho-eIF2α Ser51 (1/1,000, Cell Signaling Technology), eIF2α (1/1,000, Cell Signaling Technology), PAI-1 (1/1,000, Abcam), Rb (1/2,000, Abcam), pRb ($S^{795}$) (1/500, Abcam), HSP70 (1/10000, Stress Gene), Atg5 (1/1000, Cell Signaling Technology), tubulin (1/1,000), β-actin (1/1,000).

## De novo protein synthesis

Cells were treated with increasing concentrations of cycloheximide (Sigma) for 30 min, or fixed concentrations of AS (0.5 mM) with or without cycloheximide (0.5 μg/ml). Cells were subsequently incubated with $^{35}$S-methionine for 30 min to assess de novo global protein synthesis. Whole-cell extracts were resolved on a 10% SDS–polyacrylamide gel and stained with Coomassie blue.

## Soluble/Insoluble fractionation

Soluble and insoluble fractions were prepared as previously described [32]. Cells were resuspended in polysome buffer (20 mM Tris pH 7.4; 100 mM KCL; 0.5% NP-40; 1.25 mM $MgCl_2$ + protease inhibitor) and rocked on an orbital shaker. The soluble fraction, made up of the supernatant containing polysome buffer and soluble proteins which diffused from formed pores of the cells, was then removed and mixed with one volume of 2x Laemmli loading dye. The remaining cells were scraped in PBS and mixed with 2x Laemmli loading dye forming the insoluble fraction. Protein samples were boiled, resolved in 10% SDS–polyacrylamide gel, transferred to nitrocellulose membrane, and incubated with the indicated antibodies. Fractions enriched for SGs were analyzed using mass spectrometry at the Centre de recherche du CHU de Québec.

## PAI-1 Inhibition via TM5441

Cells induced into senescence were treated with 10 μM TM5441 (Tocris Bioscience) for 24 h daily after treatment with AS [12,13,67–69].

## Recombinant PAI-1 media supplementation

Human recombinant PAI-1 was purchased from Sigma-Aldrich (A8111-25UG) and was dissolved in DMEM (10% charcoal-stripped FBS, 1% penicillin–streptomycin) at a final concentration of 200 ng/ml. This concentration was determined using ELISA to replicate standard secretion of PAI-1 in media. 200 ng/ml of human recombinant PAI-1 was supplemented daily for each experiment after treatment with sodium arsenite or pateamine A.

## ELISA procedure

Cell media was subjected to ELISA using the PAI-1 Ready-to-use Sandwich ELISA from eBioscience. Cell media was diluted 1/50 and 100 μl was added to the micro-wells; 50 μl of biotin conjugate was added to each well for two hours, then washed, and streptavidin–HRP was added to each well and incubated for 1 h. Wells were again washed and 100 μl of prepared TMB substrate solution was added and incubated for 10 min, and 100 μl of stop solution was quickly added. Absorbances was read at 450 nm and quantified against the provided standard.

## Karyotyping of WI-38 human lung fibroblasts

Metaphase preparation and GTG banding techniques were performed according to standard cytogenetic procedures. The karyotype was described according to the International System for Human Cytogenomic Nomenclature. Ref: McGowan-Jordan J., Simons A., Schmid M. ISCN, An International System for Human Cytogenomic Nomenclature (2016). Basel, Karger.

**Expanded View** for this article is available online.

## Acknowledgements

This work is funded by a Qatar National Research Fund (QNRF) (NPRP8-457-3-101) to I.E.G, and a CIHR operating grant (MOP-89798) to I.E.G. A.O. and D.P. were funded by a scholarship received from Defi Canderel Studentship Award, McGill University, and subsequently from the The Fonds de recherche du Québec—Santé (FRQS). We would also like to thank the Advanced Bio-Imaging Facility at McGill University for their generous aid in acquiring all relevant confocal images.

## Author contributions

AO contributed to the conceptualization, conducted the investigation, validated the experimental findings, wrote the original draft, and performed the formal analysis and visualization of experimental findings. DP, XJL, JS, and SDM contributed to the investigation and validation of the senescence models, and with conceptualization, data analysis and helped edit and review the manuscript. MG and AP contributed to the interpretation of the data and furthermore reviewed and edited the manuscript. IEG conceptualized, established, and directed the execution of research goals, interpreted the data and reviewed and edited the manuscript.

## Conflict of interest

The authors declare that they have no conflict of interest.

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
