## [Review Process File · EMBO Reports]

Stress granules counteract senescence by sequestration of PAI-1

Amr Omer, Devan Patel, Xian Jin Lian, Jason Sadek, Sergio Di Marco, Arnim Pause, Myriam Gorospe, Imed Eddine Gallouzi

Review timeline:

Submission date:	28 June 2017
Editorial Decision:	28 August 2017
Revision received:	15 January 2018
Editorial Decision:	19 February 2018
Revision received:	23 February 2018
Accepted:	6 March 2018

Editor: Esther Schnapp

Transaction Report:

1st Editorial Decision

28 August 2017

Thank you for your patience, and I am very sorry for the unusual delay in the decision process of your manuscript. My colleague Martina told you that she was looking for a third opinion on your study by an expert advisor, and we have now received her/his comments, which are pasted below along with the 2 referee reports.

I am sorry to say that the evaluation of your manuscript is not a positive one. As you will see, while the referees and the advisor acknowledge that the study is potentially interesting, they also point out that it is not sufficiently conclusive, that it remains at a correlative level, and that the most important experiments would need to be performed in primary fibroblasts or vascular endothelial cells.

Given these concerns, the amount of work required to address them, the uncertain outcome of these experiments, and the fact that EMBO reports can only invite revision of papers that receive enthusiastic support from a majority of referees, I am sorry to say that we cannot offer to publish your manuscript at this stage.

However, in case you feel that you can fully address the referee concerns in a timely manner and obtain data that would considerably strengthen the message of the study, especially a causal role for PAI-1 in senescence, then we would have no objection to consider a new manuscript on the same topic in the near future. Please note that if you were to send a new manuscript this would be treated as a new submission rather than a revision and would be reviewed afresh, also with respect to the literature and the novelty of your findings at the time of resubmission.

At this stage of analysis, I am sorry to have to disappoint you. I nevertheless hope, that the referee comments will be helpful in your continued work in this area, and I thank you once more for your interest in our journal.

REFeree REPORTS

Referee #2:

This is a very interesting manuscript that advances our understanding of the mechanisms linking PAI-1 with senescence. The discovery that stress granules provide a functional site for the sequestration of an important component of the senescence-messaging system is potentially a major finding and adds to the growing body of evidence that PAI-1 is more than just a biomarker. In this original research article, Omer and colleagues investigate the impact of chronic stress on the formation of stress granules (SGs) and the molecular basis of SG-mediated suppression of cellular senescence. Authors demonstrate that i) Repeated or chronic oxidative stress inhibits the formation of SGs; ii) SGs formation in the early stage reduces the number of senescent cells; iii) SGs impair PAI-1/pRB-signaling axis; and iv) recombinant human PAI-1 treatment reverses the SG-mediated delay in senescence. Based on these observations authors conclude that chronic oxidative stress-induced SGs protect cells from undergoing senescence through recruitment of key senescence regulator PAI-1 and preventing its secretion.

General comments:

This is an important study in the context of understanding the role of SGs in prevention of cellular senescence and aging. In this well-written article authors delineated elegantly the molecular mechanism by which cells respond to chronic oxidative stresses to protect themselves from becoming senescent through induction of SGs and its molecular basis. This is a well-organized, well-planned and well-executed study with convincing data. Although this study intended to understand the significance of chronic stress-induced SGs in abrogation of cellular senescence and its implication in delayed aging process, there are few shortcomings which are as follows.

Major concerns:

Although WI38 normal lung fibroblasts were used in few experiments (Suppl. Figures 1, 2 and 8), authors used IDH4 cell line in most of the major and important experiments: Figure 1-8, Suppl. Figures 3-7 and 9-11 to understand the role of SGs in senescence. As IDH4 is dexamethasone inducible MMTV promoter driven SV40 T-antigen immortalized lung fibroblasts, this is not an ideal cell line to study senescence especially in the context of aging because biochemical and molecular characteristics of IDH4 cells are not identical to normal primary cells. It is also important to determine the chromosomal disorders in established WI38 cells if any because at higher passage this cell line may show chromosomal abnormalities.

In Page 6, We observed no differences in the expression levels of these proteins between UNT and AS treated cells (Figure 1B), suggesting that the reduced senescence seen with repeated AS treatment was not due to changes in the expression levels of several proteins known to participate in cell senescence. To draw this conclusion, authors must measure the levels of SV40 T antigen in PRO/PRE/SEN stages in Figure 1B. Additionally, to rule out the uninvolved of p53, p21 and p16 in arsenite induced SG-mediated reduction of senescence, authors should repeat the experiment using a primary culture of fibroblasts or vascular endothelial cells. In Figure 1B, the elevated levels of p53 in PRO and PRE may be simply due to presence of low levels of SV40T antigen on day 0-6 that stabilize p53. It is important to note that to reach to undetectable level of SV40T antigen in immunoblot, IDH4 cells need to grow for 14 days in the absence of dexamethasone [Ref. MCB, June 1996, 16: 2932-2939].

In order to implicate the outcome of the present study in aging-associated senescence, authors may consider to repeat few key experiments [e.g., experiments in Figure 1 and 8, Suppl. Figure 4 and 10] using normal primary cultures of fibroblasts or vascular endothelial cells. The anticipated data using primary cells will strengthen the original findings and help to draw strong conclusion that chronic stress-induced SGs contribute to alleviate chronological aging associated cellular senescence through suppression of PAI-1 secretion.

Other questions/comments:

- 1) Have the authors investigated this relationship in vivo? Have tissues from young vs old animals been examined for stress granules and the localization of PAI-1?
- 2) Small molecule antagonists of PAI-1 are now available commercially. Have the investigators

examined the effects of these drugs on cellular senescence?

3) Have the authors uncovered any evidence that other members of the SMS are sequestered in stress granules?

Referee #3:

In the submitted manuscript by Omer et al., the authors argue the stress granules (SGs) have a role in regulating senescence. Due to the pleiotropic nature of all of the factor involved (SGs), SG-inducers, senescence inducers, and SG inhibition strategies, the manuscript relies strongly on the correlation between SG formation and prevention of senescence.

Without greater mechanistic insight, the correlation between SG formation and senescence reduction is insufficiently novel. Oxidative stress does many things to the cell, including activating the unfolded protein response and other stress response pathway, which are as likely, if not moreso, to have the observed effect on senescence.

Many relevant pathways, including stress response, autophagy, and proteasome function are not examined in the manuscript - the pathways are likely to regulate upstream events, as well as the exocytic pathway, and are therefore suspects for contributing to the delay in senescence.

Interpreting the mass spec results is tricky because they are based on the assumption that the relevant factors will be in the insoluble fraction. There is evidence in the field that by the time SGs are in an insoluble fraction, they are less representative of the functional SG state, and have transitioned more towards an aberrant aggregated state, which would then introduce a lot of potential for artifactual localization of misfolded proteins and DRiPs.

Advisor's comments:

Upon reading this MS (which I found interesting) and the comments provided by the two referees, I have to say that I have some of the same concerns as referee #3: that is that the link between SG formation and a PAI-1-dependent effect on senescence is mainly based on correlations and no clear cause-and-effect is shown in the paper. In fact, I believe referee #2 has some of the same concerns in stating "Small molecule antagonists of PAI-1 are now available commercially. Have the investigators examined the effects of these drugs on cellular senescence?". If, as the authors suggest, SG formation is sequestering PAI-1 to inhibit senescence then antagonists of PAI-1 should mimic this response (this is important as it is not known if PAI-1 inhibition on its own (with the cell line studied) reduces the cell's ability to enter senescence. An even stronger evidence for their model would be to ectopically over-activate PAI-1 (e.g. by overexpression) to see if this overcomes the inhibitory effect of SG formation on senescence.

Also, I feel that referee #2's unease about the cell line used (in most of the experiments) and suggestion of using other ones (perhaps also in vivo verification) are very valid concerns.

Also 2, since the study does not include any data on aging it would be better not to speculate so much on this.

In summary, an interesting paper that, in my view, doesn't quite reach the level of mechanistic insights (yet) that is customarily seen in EMBO R.

1st Revision - authors' response

15 January 2018

Referee #2:

This is a very interesting manuscript that advances our understanding of the mechanisms linking PAI-1 with senescence. The discovery that stress granules provide a functional site for the sequestration of an important component of the senescence-messaging system is potentially a major finding and adds to the growing body of evidence that PAI-1 is more than just a biomarker. In this

original research article, Omer and colleagues investigate the impact of chronic stress on the formation of stress granules (SGs) and the molecular basis of SG-mediated suppression of cellular senescence. Authors demonstrate that i) Repeated or chronic oxidative stress inhibits the formation of SGs; ii) SGs formation in the early stage reduces the number of senescent cells; iii) SGs impair PAI-1/pRB-signaling axis; and iv) recombinant human PAI-1 treatment reverses the SG-mediated delay in senescence. Based on these observations authors conclude that chronic oxidative stress-induced SGs protect cells from undergoing senescence through recruitment of key senescence regulator PAI-1 and preventing its secretion.

General comments:

This is an important study in the context of understanding the role of SGs in prevention of cellular senescence and aging. In this well-written article authors delineated elegantly the molecular mechanism by which cells respond to chronic oxidative stresses to protect themselves from becoming senescent through induction of SGs and its molecular basis. This is a well-organized, well-planned and well-executed study with convincing data. Although this study intended to understand the significance of chronic stress-induced SGs in abrogation of cellular senescence and its implication in delayed aging process, there are few shortcomings which are as follows.

Major points:

I. Although WI38 normal lung fibroblasts were used in few experiments (Suppl. Figures 1, 2 and 8), authors used IDH4 cell line in most of the major and important experiments: Figure 1-8, Suppl. Figures 3-7 and 9-11 to understand the role of SGs in senescence. As IDH4 is dexamethasone inducible MMTV promoter driven SV40 T-antigen immortalized lung fibroblasts, this is not an ideal cell line to study senescence especially in the context of aging because biochemical and molecular characteristics of IDH4 cells are not identical to normal primary cell.

We agree with the reviewer that the biochemical and molecular characteristics of IDH4 cells are not identical to those of primary cells such as WI-38 fibroblasts. Although the majority of the important data contained in our original manuscript were obtained using IDH4 cells, in the revised manuscript we have included new data demonstrating that the conclusions drawn from these experiments were also found in normal WI-38 fibroblasts. Evidence that all the major experiments performed using IDH4 cells were also replicated using WI38 cells can be found in the revised manuscript, Supplemental Figs. 2, 5, 13, 15, and 17. These results provide further support to our conclusions that the chronic stress-induced recruitment of PAI-1 to stress granules protected cells from becoming senescent.

I.a. It is also important to determine the chromosomal disorders in established WI38 cells if any because at higher passage this cell line may show chromosomal abnormalities.

As suggested by the reviewer, we have assessed whether WI-38 fibroblasts have chromosomal abnormalities at the passages (12 to 25) used in our publication. Karyotyping of these cells at the time of usage has revealed that they do not possess any chromosomal abnormalities (see Supp. Fig. 1b).

I.b. In Page 6, We observed no differences in the expression levels of these proteins between UNT and AS treated cells (Figure 1B), suggesting that the reduced senescence seen with repeated AS treatment was not due to changes in the expression levels of several proteins known to participate in cell senescence. To draw this conclusion, authors must measure the levels of SV40 T antigen in PRO/PRE/SEN stages in Figure 1B... In Figure 1B, the elevated levels of p53 in PRO and PRE may be simply due to presence of low levels of SV40T antigen on day 0-6 that stabilize p53. It is important to note that to reach to undetectable level of SV40T antigen in immunoblot, IDH4 cells need to grow for 14 days in the absence of dexamethasone [Ref. MCB, June 1996, 16: 2932-2939]. **We thank the reviewer for suggesting this experiment. We have assessed the expression levels of SV40 T-antigen during the senescence process by Western Blot analysis. Our results indicate that the expression of T-antigen decreases to undetectable levels after the induction of senescence (removal of dexamethasone and addition of charcoal-stripped FBS) (Supp. Fig. 1A). Furthermore, the expression of SV40 was not affected by AS treatment upon induction of this process.**

I.c. Additionally, to rule out the uninvolved of p53, p21 and p16 in arsenite induced SG-mediated reduction of senescence, authors should repeat the experiment using a primary culture of fibroblasts or vascular endothelial cells... In order to implicate the outcome of the present study in aging-associated senescence, authors may consider to repeat few key experiments [e.g., experiments in Figure 1 and 8, Suppl. Figure 4 and 10] using normal primary cultures of fibroblasts or vascular endothelial cells. The anticipated data using primary cells will strengthen the original findings and help to draw strong conclusion that chronic stress-induced SGs contribute to alleviate chronological aging associated cellular senescence through suppression of PAI-1 secretion.

As mentioned above, all of the important data obtained using IDH4 cells in our original manuscript were replicated using primary human lung WI-38 fibroblasts (Supp. Figs. 2, 5, 13, 15, and 17). The inclusion of these data helped to strengthen our conclusions that stress-induced SGs protect cells from senescent by suppressing the secretion of PAI-1.

Other questions/comments:

1. Have the authors investigated this relationship in vivo? Have tissues from young vs old animals been examined for stress granules and the localization of PAI-1?

We thank the reviewer for this comment. Addressing the relationship between the SG-mediated repression of PAI-1 secretion and senescence in vivo is very important and would of course strengthen our manuscript. Although we have recently begun to investigate this relationship in vivo, the results generated are still in the early stages, and, in our opinion are beyond the scope of this present study. However, we have a long-term commitment to performing these studies and we hope to complete them within a year or so.

2. Small molecule antagonists of PAI-1 are now available commercially. Have the investigators examined the effects of these drugs on cellular senescence?

We appreciate the reviewer's suggestion. We have assessed the effect of TM5441, a commercially available PAI-1 inhibitor, on cellular senescence. Our data demonstrate that inhibition of PAI-1 using TM5441 delays senescence similarly to the early formation of SGs (Fig. 7C). Furthermore, co-treatment of cells with TM5441 and AS did not further impact senescence, suggesting that the SG-mediated decrease in senescence is indeed dependent on its effect on PAI-1 secretion.

3. Have the authors uncovered any evidence that other members of the SMS are sequestered in stress granules?

We thank the reviewer for this comment. So far, we do not have any evidence that other members of the SASP family go to stress granules. In addition, our mass spectrometry analysis (Supp. Table 1 and 2) suggest that PAI-1 is the only member of this family that is enriched in the insoluble fraction of AS treated extracts.

Referee #3:

In the submitted manuscript by Omer et al., the authors argue the stress granules (SGs) have a role in regulating senescence. Due to the pleiotropic nature of all of the factor involved (SGs), SG-inducers, senescence inducers, and SG inhibition strategies, the manuscript relies strongly on the correlation between SG formation and prevention of senescence.

General comments:

Without greater mechanistic insight, the correlation between SG formation and senescence reduction is insufficiently novel. Oxidative stress does many things to the cell, including activating the unfolded protein response and other stress response pathway, which are as likely, if not more so, to have the observed effect on senescence.

We agree with the reviewer that oxidative stress may activate other pathways that could be involved in the observed delay in senescence. We addressed this concern as described below.

1. Many relevant pathways, including stress response, autophagy, and proteasome function are not examined in the manuscript the pathways are likely to regulate upstream events, as well as the exocytic pathway, and are therefore suspects for contributing to the delay in senescence.

We agree with the reviewer that the AS-mediated effect on senescence may also be due to the activation of additional processes including the stress response, autophagy, and proteasome function in our cells. In order to address these possibilities, we systematically inhibited these processes and assessed the impact of these interventions on senescence (Supp. Fig. 9, 10). We used siRNAs to knock down HSP70 or Atg5, core components of the stress response and autophagy pathway respectively, and used MG132 to inhibit the proteasome. Our experiments indicate that the effect of AS on senescence are not due to the activation or inhibition of these pathways. Additionally, since the effect of AS on this process was also observed when cells were treated with Pateamine A (which induces the formation of SGs in a manner distinct from AS) our data support the conclusion that SGs are major drivers of the AS-mediated decrease in senescence.

2. Interpreting the mass spec results is tricky because they are based on the assumption that the relevant factors will be in the insoluble fraction. There is evidence in the field that by the time SGs are in an insoluble fraction, they are less representative of the functional SG state, and have transitioned more towards an aberrant aggregated state, which would then introduce a lot of potential for artifactual localization of misfolded proteins and DRiPs.

We agree with the reviewer that interpretation of our Mass Spectrometry (MS) data is indeed tricky and that the soluble fraction contains factors that are not part of SGs. However, our subsequent data confirmed the recruitment of PAI-1 to SGs. This result was obtained by immunofluorescence in both IDH4 and WI-38 fibroblasts treated with 2 different inducers of SGs, AS and Pateamine A. These data strengthen our conclusion that PAI-1 is sequestered to SGs during senescence in these cells.

Advisor:

General comments:

Upon reading this MS (which I found interesting) and the comments provided by the two referees, I have to say that I have some of the same concerns as referee #3: that is that the link between SG formation and a PAI-1-dependent effect on senescence is mainly based on correlations and no clear cause-and-effect is shown in the paper.

1. In fact, I believe referee #2 has some of the same concerns in stating "Small molecule antagonists of PAI-1 are now available commercially. Have the investigators examined the effects of these drugs on cellular senescence?". If, as the authors suggest, SG formation is sequestering PAI-1 to inhibit senescence then antagonists of PAI-1 should mimic this response (this is important as it is not known if PAI-1 inhibition on its own (with the cell line studied) reduces the cell's ability to enter senescence...). In summary, an interesting paper that, in my view, doesn't quite reach the level of mechanistic insights (yet) that is customarily seen in EMBO R.

We appreciate this comment. These concerns were addressed in our response to the comments from reviewers 2 and 3.

2. An even stronger evidence for their model would be to ectopically over-activate PAI-1 (e.g. by overexpression) to see if this overcomes the inhibitory effect of SG formation on senescence.

We thank the advisor for suggesting this experiment. We overexpressed PAI-1 ectopically in IDH4 cells and found that it negated the inhibitory effect of SG formation on senescence (Fig. 7C). This experiment further suggests that PAI-1 recruitment to SGs impairs senescence under normal conditions.

3. Also, I feel that referee #2's unease about the cell line used (in most of the experiments) and suggestion of using other ones (perhaps also in vivo verification) are very valid concerns. **These concerns were addressed as described above to respond to the concerns of reviewer 2.**

4. Also 2, since the study does not include any data on aging it would be better not to speculate so much on this.

We agree with the reviewer and have amended the text accordingly.

Thank you for the submission of your revised manuscript. We have now received the enclosed report from referee 3 whom I asked to assess your response to all remaining referee concerns. I am glad to tell you that referee 3 supports the publication of your manuscript now. A few more minor changes only need to be made before we can proceed with the official acceptance of your manuscript.

Please send us a completed author checklist that can be found here:

<http://embor.embopress.org/authorguide#revision>

The completed author checklist will also be part of our transparent peer-review process file (RPF).

Figures 1A, 3C, 4B, 5B, 7B,C, 8, EV3, EV4B, S2, S5, S7B, S8B,D, S9B, S12, S13 need scale bars.

Figures 3A, 8B, EV1, EV3, EV4B, S6B, S8B,D, S9B, S11 and S12 all state $n=2$ but error bars and p-values are calculated, which is not possible. At least three independently performed experiments are required to calculate statistics. Please either repeat the experiments one more time, or show the single data points of both experiments along with their mean.

Figures S5A,B and S6A need data info for all panels and the microscopy images of S5A are out of focus.

I attach a word file with comments on the figure legends to this email, please use this file and send us back a corrected word file.

The heading "experimental procedures" needs to be changed to Materials and Methods.

Please send us a running title, up to 5 keywords and a conflict of interest statement.

In the manuscript text, the figures 4ab & 5ab and 7abc panels need to be called out.

In Figs 6E, EV2 & S10 below the expanded box there are two boxes with arrows. It is unclear what these boxes show, please explain this in the figure legends and/or add more information to the figures.

In Fig 8C it is unclear what the merged images show, please explain.

I would like to suggest a few minor changes to the abstract that needs to be written in present tense:

Cellular senescence is a physiological response by which an organism halts the proliferation of potentially harmful and damaged cells. However, the accumulation of senescent cells over time can become deleterious leading to diseases and physiologic decline. Our data reveal a novel interplay between senescence and the stress response that affects both the progression of senescence and the behavior of senescent cells. We show that constitutive exposure to stress induces the formation of stress granules (SGs) in proliferative and pre-senescent cells, but not in fully senescent cells. SG assembly alone is sufficient to decrease the number of senescent cells without affecting the expression of bona fide senescence markers. SGs-mediated inhibition of senescence is associated with the recruitment of the plasminogen activator inhibitor-1 (PAI-1), a known promoter of senescence, to these entities. PAI-1 localization to SGs decreases the translocation of cyclin D1 to

the nucleus, suppressing RB phosphorylation and maintaining a proliferative, non-senescent state. Together, our data indicate that SGs may be targets of intervention to modulate senescence in order to impair or prevent its deleterious effects.

Please send us the synopsis image as image file (eg tif, jpg, photoshop) as we cannot work with the pdf file, and the short summary and bullet points as word file.

REFEREE REPORT

Referee #3:

The authors have improved their manuscript and I think that the idea is sufficiently interesting to merit publication.

2nd Revision - authors' response

23 February 2018

- **We accept to take part of the "EMBO publication's Transparent Editorial Process"**

2) Figures 1A, 3C, 4B, 5B, 7B,C, 8, EV3, EV4B, S2, S5, S7B, S8B,D, S9B, S12, S13 need scale bars.

- **All images mentioned above have had scale bars added to them and corresponding figure legends have been amended to present correct scaling.**

3) Figures 3A, 8B, EV1, EV3, EV4B, S6B, S8B,D, S9B, S11 and S12 all state n=2 but error bars and p-values are calculated, which is not possible. At least three independently performed experiments are required to calculate statistics. Please either repeat the experiments one more time, or show the single data points of both experiments along with their mean.

- **Bar graphs for Figures 8B, EV1, EV3, EV4B, S6B, S6D, and S12 have been amended. New graphs showing single data points and means have been added.**
- **Figures 3A, S9B and S11 were repeated and corresponding graphs and statistics are shown.**

4) Figures S5A,B and S6A need data info for all panels and the microscopy images of S5A are out of focus.

- **Data information for S5A,B and S6A have been added to their corresponding figure legends.**
- **Images for S5A were retaken from the same experiment replacing the corresponding figure.**

5) I attach a word file with comments on the figure legends to this email, please use this file and send us back a corrected word file.

- **The changes to figure legends suggested by the editor have all been made.**
- **The suggested changes to the text were also accepted.**

6) The heading "experimental procedures" needs to be changed to Materials and Methods.

- **The heading has been replaced as suggested.**

- 7) Please send us a running title, up to 5 keywords and a conflict of interest statement.
- **Running title has been added to the cover page of the main text (page 1)**
 - **Key words have been included below the abstract (page 2) of the main text word file.**
 - **A conflict of interest statement has been included below the acknowledgement section (page 16) of main text word file.**
- 8) In the manuscript text, the figures 4ab & 5ab and 7abc panels need to be called out.
- **The text has been amended to reflect this suggestion.**
- 9) In Figs 6E, EV2 & S10 below the expanded box there are two boxes with arrows. It is unclear what these boxes show, please explain this in the figure legends and/or add more information to the figures.
- **An explanation for the boxes and arrows have included in the corresponding figure legends. The statement used is as follows: The red square in the AS panel represents the area that was expanded and is shown on the right. The two panels below the expanded box show individual staining for PAI-1 (red) and FMRP (green). Arrows indicate examples of co-localized PAI-1 and FMRP in the same foci. Scale bars, 50 μ m.**
- 10) In Fig 8C it is unclear what the merged images show, please explain.
- **The merged images have been more clearly explained in its corresponding figure legend. The merged images correspond to DAPI and Cyclin D1 co-staining of the same slide.**
- 11) I would like to suggest a few minor changes to the abstract that needs to be written in present tense:
- **Thank you so much for the great changes you made to our abstract. We accepted them all.**

YOU MUST COMPLETE ALL CELLS WITH A PINK BACKGROUND ↓
PLEASE NOTE THAT THIS CHECKLIST WILL BE PUBLISHED ALONGSIDE YOUR PAPER

Corresponding Author Name: Imed Gallouzi
Journal Submitted to: EMBO Report
Manuscript Number: EMBOR-2017-44722V2